# Tensor Switching Networks

**Chuan-Yung Tsai**\*, **Andrew Saxe**\*, **David Cox**
Center for Brain Science, Harvard University, Cambridge, MA 02138
{chuanyungtsai,asaxe,davidcox}@fas.harvard.edu

## Abstract

We present a novel neural network algorithm, the Tensor Switching (TS) network, which generalizes the Rectified Linear Unit (ReLU) nonlinearity to tensor-valued hidden units. The TS network copies its entire input vector to different locations in an expanded representation, with the location determined by its hidden unit activity. In this way, even a simple linear readout from the TS representation can implement a highly expressive deep-network-like function. The TS network hence avoids the vanishing gradient problem by construction, at the cost of larger representation size. We develop several methods to train the TS network, including equivalent kernels for infinitely wide and deep TS networks, a one-pass linear learning algorithm, and two backpropagation-inspired representation learning algorithms. Our experimental results demonstrate that the TS network is indeed more expressive and consistently learns faster than standard ReLU networks.

## 1 Introduction

Deep networks [1, 2] continue to post impressive successes in a wide range of tasks, and the Rectified Linear Unit (ReLU) [3, 4] is arguably the most used simple nonlinearity. In this work we develop a novel deep learning algorithm, the Tensor Switching (TS) network, which generalizes the ReLU such that each hidden unit conveys a tensor, instead of scalar, yielding a more expressive model. Like the ReLU network, the TS network is a linear function of its input, conditioned on the activation pattern of its hidden units. By separating the decision to activate from the analysis performed when active, even a linear classifier can reach back across all layers to the input of the TS network, implementing a deep-network-like function while avoiding the vanishing gradient problem [5], which can otherwise significantly slow down learning in deep networks. The trade-off is the representation size.

We exploit the properties of TS networks to develop several methods suitable for learning in different scaling regimes, including their equivalent kernels for SVMs on small to medium datasets, a one-pass linear learning algorithm which visits each data point only once for use with very large but simpler datasets, and two backpropagation-inspired representation learning algorithms for more generic use. Our experimental results show that TS networks are indeed more expressive and consistently learn faster than standard ReLU networks.

Related work is briefly summarized as follows. With respect to improving the nonlinearities, the idea of severing activation and analysis weights (or having multiple sets of weights) in each hidden layer has been studied in [6, 7, 8]. Reordering activation and analysis is proposed by [9]. On tackling the vanishing gradient problem, tensor methods are used by [10] to train single-hidden-layer networks. Convex learning and inference in various deep architectures can be found in [11, 12, 13] too. Finally, conditional linearity of deep ReLU networks is also used by [14], mainly to analyze their performance. In comparison, the TS network does not simply reorder or sever activation and analysis within each hidden layer. Instead, it is a cross-layer generalization of these concepts, which can be applied with most of the recent deep learning architectures [15, 9], not only to increase their expressiveness, but also to help avoiding the vanishing gradient problem (see Sec. 2.3).

---

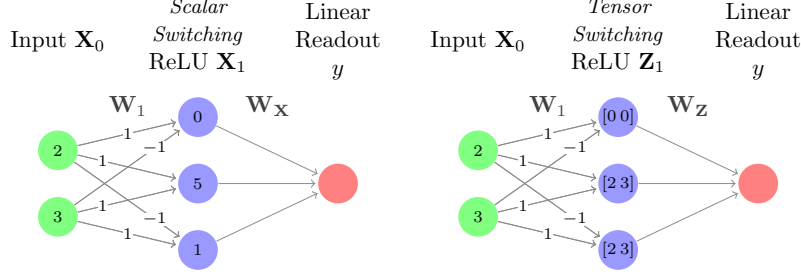

Figure 1: (Left) A single-hidden-layer standard (*i.e.* Scalar Switching) ReLU network. (Right) A single-hidden-layer Tensor Switching ReLU network, where each hidden unit conveys a vector of activities—inactive units (top-most unit) convey a vector of zeros while active units (bottom two units) convey a copy of their input.

## 2   Tensor Switching Networks

In the following we first construct the definition of shallow (single-hidden-layer) TS networks, then generalize the definition to deep TS networks, and finally describe their qualitative properties. For simplicity, we only show fully-connected architectures using the ReLU nonlinearity. However, other popular nonlinearities, *e.g.* max pooling and maxout [16], in addition to ReLU, are also supported in both fully-connected and convolutional architectures.

### 2.1   Shallow TS Networks

The TS-ReLU network is a generalization of standard ReLU networks that permits each hidden unit to convey an entire tensor of activity (see Fig. 1). To describe it, we build up from the standard ReLU network. Consider a ReLU layer with weight matrix $\mathbf{W}_1 \in \mathbb{R}^{n_1 \times n_0}$ responding to an input vector $\mathbf{X}_0 \in \mathbb{R}^{n_0}$. The resulting hidden activity $\mathbf{X}_1 \in \mathbb{R}^{n_1}$ of this layer is $\mathbf{X}_1 = \max\left(\mathbf{0}^{n_1}, \mathbf{W}_1 \mathbf{X}_0\right) = H\left(\mathbf{W}_1 \mathbf{X}_0\right) \circ \left(\mathbf{W}_1 \mathbf{X}_0\right)$ where $H$ is the Heaviside step function, and $\circ$ denotes elementwise product. The rightmost equation splits apart each hidden unit's decision to activate, represented by the term $H\left(\mathbf{W}_1 \mathbf{X}_0\right)$, from the information (*i.e.* result of analysis) it conveys when active, denoted by $\mathbf{W}_1 \mathbf{X}_0$. We then go one step further to rewrite $\mathbf{X}_1$ as

$$\mathbf{X}_1 = \left( \underbrace{H\left(\mathbf{W}_1 \mathbf{X}_0\right) \otimes \mathbf{X}_0}_{\mathbf{Z}_1} \odot \mathbf{W}_1 \right) \times \mathbf{1}^{n_0}, \tag{1}$$

where we have made use of the following tensor operations: vector-tensor cross product $\mathbf{C} = \mathbf{A} \otimes \mathbf{B} \implies c_{i,j,k,\ldots} = a_i b_{j,k,\ldots}$, tensor-matrix Hadamard product $\mathbf{C} = \mathbf{A} \odot \mathbf{B} \implies c_{\ldots,j,i} = a_{\ldots,j,i} b_{j,i}$ and tensor summative reduction $\mathbf{C} = \mathbf{A} \times \mathbf{1}^n \implies c_{\ldots,k,j} = \sum_{i=1}^{n} a_{\ldots,k,j,i}$. In (1), the input vector $\mathbf{X}_0$ is first expanded into a new matrix representation $\mathbf{Z}_1 \in \mathbb{R}^{n_1 \times n_0}$ with one row per hidden unit. If a hidden unit is active, the input vector $\mathbf{X}_0$ is copied to the corresponding row. Otherwise, the row is filled with zeros. Finally, this expanded representation $\mathbf{Z}_1$ is collapsed back by projection onto $\mathbf{W}_1$.

The central idea behind the TS-ReLU network is to learn a linear classifier directly from the rich, expanded representation $\mathbf{Z}_1$, rather than collapsing it back to the lower dimensional $\mathbf{X}_1$. That is, in a standard ReLU network, the hidden layer activity $\mathbf{X}_1$ is sent through a linear classifier $f_{\mathbf{X}}\left(\mathbf{W}_{\mathbf{X}} \mathbf{X}_1\right)$ trained to minimize some loss function $\mathcal{L}_{\mathbf{X}}\left(f_{\mathbf{X}}\right)$. In the TS-ReLU network, by contrast, the expanded representation $\mathbf{Z}_1$ is sent to a linear classifier $f_{\mathbf{Z}}\left(\mathbf{W}_{\mathbf{Z}} \operatorname{vec}\left(\mathbf{Z}_1\right)\right)$ with loss function $\mathcal{L}_{\mathbf{Z}}\left(f_{\mathbf{Z}}\right)$. Each TS-ReLU neuron thus transmits a vector of activities (a row of $\mathbf{Z}_1$), compared to a standard ReLU neuron that transmits a single scalar (see Fig. 1). Because of this difference, in the following we call the standard ReLU network a Scalar Switching ReLU (SS-ReLU) network.

### 2.2   Deep TS Networks

The construction given above generalizes readily to deeper networks. Define a nonlinear expansion operation as $\mathbf{X} \oplus \mathbf{W} = H\left(\mathbf{W}\mathbf{X}\right) \otimes \mathbf{X}$ and linear contraction operation as $\mathbf{Z} \ominus \mathbf{W} = \left(\mathbf{Z} \odot \mathbf{W}\right) \times \mathbf{1}^n$, such that (1) becomes $\mathbf{X}_l = \left(\left(\mathbf{X}_{l-1} \oplus \mathbf{W}_l\right) \odot \mathbf{W}_l\right) \times \mathbf{1}^{n_{l-1}} = \mathbf{X}_{l-1} \oplus \mathbf{W}_l \ominus \mathbf{W}_l$ for a given layer $l$

with $\mathbf{X}_l \in \mathbb{R}^{n_l}$ and $\mathbf{W}_l \in \mathbb{R}^{n_l \times n_{l-1}}$. A deep SS-ReLU network with $L$ layers may then be expressed as a sequence of alternating expansion and contraction steps,

$$\mathbf{X}_L = \mathbf{X}_0 \oplus \mathbf{W}_1 \ominus \mathbf{W}_1 \cdots \oplus \mathbf{W}_L \ominus \mathbf{W}_L. \tag{2}$$

To obtain the deep TS-ReLU network, we further define the ternary expansion operation $\mathbf{Z} \oplus_{\mathbf{X}} \mathbf{W} = H(\mathbf{WX}) \otimes \mathbf{Z}$, such that the decision to activate is based on the SS-ReLU variables $\mathbf{X}$, but the entire tensor $\mathbf{Z}$ is transmitted when the associated hidden unit is active. Let $\mathbf{Z}_0 = \mathbf{X}_0$. The $l$-th layer activity tensor of a TS network can then be written as $\mathbf{Z}_l = H(\mathbf{W}_l \mathbf{X}_{l-1}) \otimes \mathbf{Z}_{l-1} = \mathbf{Z}_{l-1} \oplus_{\mathbf{X}_{l-1}} \mathbf{W}_l \in \mathbb{R}^{n_l \times n_{l-1} \times \cdots \times n_0}$. Thus compared to a deep SS-ReLU network, a deep TS-ReLU network simply omits the contraction stages,

$$\mathbf{Z}_L = \mathbf{Z}_0 \oplus_{\mathbf{X}_0} \mathbf{W}_1 \cdots \oplus_{\mathbf{X}_{L-1}} \mathbf{W}_L. \tag{3}$$

Because there are no contraction steps, the order of $\mathbf{Z}_l \in \mathbb{R}^{n_l \times n_{l-1} \times \cdots \times n_0}$ grows with depth, adding an additional dimension for each layer. One interpretation of this scheme is that, if a hidden unit at layer $l$ is active, the entire tensor $\mathbf{Z}_{l-1}$ is copied to the appropriate position in $\mathbf{Z}_l$.[1] Otherwise a tensor of zeros is copied. Another equivalent interpretation is that the input vector $\mathbf{X}_0$ is copied to a given position $\mathbf{Z}_l(i, j, \ldots, k, :)$ only if hidden units $i, j, \ldots, k$ at layers $l, l-1, \ldots, 1$ respectively are all active. Otherwise, $\mathbf{Z}_l(i, j, \ldots, k, :) = \mathbf{0}^{n_0}$. Hence activity propagation in the deep TS-ReLU network preserves the layered structure of a deep SS-ReLU network, in which a chain of hidden units across layers must activate for activity to propagate from input to output.

## 2.3 Properties

The TS network decouples a hidden unit's decision to activate (as encoded by the activation weights $\{\mathbf{W}_l\}$) from the analysis performed on the input when the unit is active (as encoded by the analysis weights $\mathbf{W}_{\mathbf{Z}}$). This distinguishing feature leads to the following 3 properties.

**Cross-layer analysis.** Since the TS representation preserves the layered structure of a deep network and offers direct access to the entire input (parcellated by the activated hidden units), a simple linear readout can effectively reach back across layers to the input and thus implicitly learns analysis weights for all layers at one time in $\mathbf{W}_{\mathbf{Z}}$. Therefore it avoids the vanishing gradient problem by construction.[2]

**Error-correcting analysis.** As activation and analysis are severed, a careful selection of the analysis weights can "clean up" a certain amount of inexactitude in the choice to activate, *e.g.* from noisy or even random activation weights. While for the SS network, bad activation also implies bad analysis.

**Fine-grained analysis.** To see this, we consider single-hidden-layer TS and SS networks with just one hidden unit. The TS unit, when active, conveys the entire input vector, and hence any full-rank linear map from input to output may be implemented. The SS unit, when active, conveys just a single scalar, and hence can only implement a rank-1 linear map between input and output. By choosing the right analysis weights, a TS network can always implement an SS network,[3] but not *vice versa*. As such, it clearly has greater modeling capacity for a fixed number of hidden units.

Although the TS representation is highly expressive, it comes at the cost of an exponential increase in the size of its representation with depth, *i.e.* $\prod_l n_l$. This renders TS networks of substantial width and depth very challenging (except as kernels). But as we will show, the expressiveness permits TS networks to perform fairly well without having to be extremely wide and deep, and often noticeably better than SS networks of the same sizes. Also, TS networks of useful sizes still can be implemented with reasonable computing resources, especially when combined with techniques in Sec. 4.3.

## 3 Equivalent Kernels

In this section we derive equivalent kernels for TS-ReLU networks with arbitrary depth and an infinite number of hidden units at each layer, with the aim of providing theoretical insight into how TS-ReLU is analytically different from SS-ReLU. These kernels represent the extreme of infinite (but unlearned) features, and might be used in SVM on datasets of small to medium sizes.

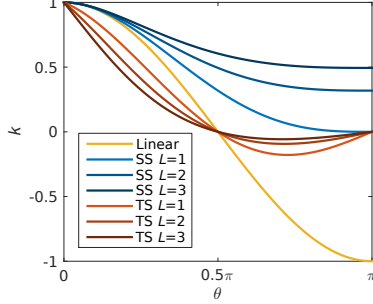

Figure 2: Equivalent kernels as a function of the angle between unit-length vectors $\mathbf{x}$ and $\mathbf{y}$. The deep SS-ReLU kernel converges to $1$ everywhere as $L \to \infty$, while the deep TS-ReLU kernel converges to $1$ at the origin and $0$ everywhere else.

Consider a single-hidden-layer TS-ReLU network with $n_1$ hidden units in which each element of the activation weight matrix $\mathbf{W}_1 \in \mathbb{R}^{n_1 \times n_0}$ is i.i.d. zero mean Gaussian with arbitrary standard deviation $\sigma$. The infinite-width random TS-ReLU kernel between two vectors $\mathbf{x}, \mathbf{y} \in \mathbb{R}^{n_0}$ is the dot product between their expanded representations (scaled by $\sqrt{2/n_1}$ for convenience) in the limit of infinite hidden units, $k_1^{\text{TS}}(\mathbf{x}, \mathbf{y}) = \lim_{n_1 \to \infty} \text{vec}\left(\sqrt{2/n_1}\,\mathbf{x} \oplus \mathbf{W}_1\right)^{\mathsf{T}} \text{vec}\left(\sqrt{2/n_1}\,\mathbf{y} \oplus \mathbf{W}_1\right) = 2\,\mathbb{E}\left[H\left(\mathbf{w}^{\mathsf{T}}\mathbf{x}\right) H\left(\mathbf{w}^{\mathsf{T}}\mathbf{y}\right)\right]\mathbf{x}^{\mathsf{T}}\mathbf{y}$, where $\mathbf{w} \sim \mathcal{N}\left(\mathbf{0}, \sigma^2 \mathbf{I}\right)$ is a $n_0$-dimensional random Gaussian vector. The expectation is the probability that a randomly chosen vector $\mathbf{w}$ lies within 90 degrees of both $\mathbf{x}$ and $\mathbf{y}$. Because $\mathbf{w}$ is drawn from an isotropic Gaussian, if $\mathbf{x}$ and $\mathbf{y}$ differ by an angle $\theta$, then only the fraction $\frac{\pi - \theta}{2\pi}$ of randomly drawn $\mathbf{w}$ will be within 90 degrees of both, yielding the equivalent kernel of a single-hidden-layer infinite-width random TS-ReLU network given in (5).[4]

$$k_1^{\text{SS}}(\mathbf{x}, \mathbf{y}) = \bar{k}^{\text{SS}}(\theta)\,\mathbf{x}^{\mathsf{T}}\mathbf{y} = \left(1 - \frac{\tan\theta - \theta}{\pi}\right)\mathbf{x}^{\mathsf{T}}\mathbf{y} \tag{4}$$

$$k_1^{\text{TS}}(\mathbf{x}, \mathbf{y}) = \bar{k}^{\text{TS}}(\theta)\,\mathbf{x}^{\mathsf{T}}\mathbf{y} = \left(1 - \frac{\theta}{\pi}\right)\mathbf{x}^{\mathsf{T}}\mathbf{y} \tag{5}$$

Figure 2 compares (5) against the linear kernel and the single-hidden-layer infinite-width random SS-ReLU kernel (4) from [20] (see Linear, TS $L = 1$ and SS $L = 1$). It has two important qualitative features. First, it has discontinuous derivative at $\theta = 0$, and hence a much sharper peak than the other kernels.[5] Intuitively this means that a very close match counts for much more than a moderately close match. Second, unlike the SS-ReLU kernel which is non-negative everywhere, the TS-ReLU kernel still has a negative lobe, though it is substantially reduced relative to the linear kernel. Intuitively this means that being dissimilar to a support vector can provide evidence against a particular classification, but this negative evidence is much weaker than in a standard linear kernel.

To derive kernels for deeper TS-ReLU networks, we need to consider the deeper SS-ReLU kernels as well, since its activation and analysis are severed, and the activation instead depends on its SS-ReLU counterpart. Based upon the recursive formulation from [20], first we define the zeroth-layer kernel $k_0^{\bullet}(\mathbf{x}, \mathbf{y}) = \mathbf{x}^{\mathsf{T}}\mathbf{y}$ and the generalized angle $\theta_l^{\bullet} = \cos^{-1}\left(k_l^{\bullet}(\mathbf{x}, \mathbf{y})/\sqrt{k_l^{\bullet}(\mathbf{x}, \mathbf{x})\,k_l^{\bullet}(\mathbf{y}, \mathbf{y})}\right)$, where $\bullet$ denotes SS or TS. Then we can easily get $k_{l+1}^{\text{SS}}(\mathbf{x}, \mathbf{y}) = \bar{k}^{\text{SS}}(\theta_l^{\text{SS}})\,k_l^{\text{SS}}(\mathbf{x}, \mathbf{y})$,[6] and $k_{l+1}^{\text{TS}}(\mathbf{x}, \mathbf{y}) = \bar{k}^{\text{TS}}(\theta_l^{\text{SS}})\,k_l^{\text{TS}}(\mathbf{x}, \mathbf{y})$, where $\bar{k}^{\bullet}$ follows (4) or (5) accordingly.

Figure 2 also plots the deep TS-ReLU and SS-ReLU kernels as a function of depth. The shape of these kernels reveals sharply divergent behavior between the TS and SS networks. As depth increases, the equivalent kernel of the TS network falls off ever more rapidly as the angle between input vectors increases. This means that vectors must be an ever closer match to retain a high kernel value. As argued earlier, this highlights the ability of the TS network to pick up on and amplify small differences between inputs, resulting in a quasi-nearest-neighbor behavior. In contrast, the equivalent kernel of the SS network limits to one as depth increases. Thus, rather than amplifying small differences, it collapses them with depth such that even very dissimilar vectors receive high kernel values.

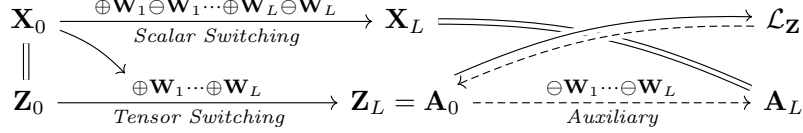

Figure 3: Inverted backpropagation learning flowchart, where $\rightarrow$ denotes signal flow, $\dashrightarrow$ denotes pseudo gradient flow, and $=$ denotes equivalence. (Top row) The SS pathway. (Bottom row) The TS and auxiliary pathways, where $\mathbf{Z}_l$'s are related by nonlinear expansions, and $\mathbf{A}_l$'s are related by linear contractions. The resulting $\mathbf{A}_L$ is equivalent to the alternating expansion and contraction in the SS pathway that yields $\mathbf{X}_L$.

## 4 Learning Algorithms

In the following we present 3 learning algorithms suitable for different scenarios. One-pass ridge regression in Sec. 4.1 learns only the linear readout (*i.e.* analysis weights $\mathbf{W_Z}$), leaving the hidden-layer representations (*i.e.* activation weights $\{\mathbf{W}_l\}$) random, hence it is convex and exactly solvable. Inverted backpropagation in Sec. 4.2 learns both analysis and activation weights. Linear Rotation-Compression in Sec. 4.3 also learns both weights, but learns activation weights in an indirect way.

### 4.1 Linear Readout Learning via One-pass Ridge Regression

In this scheme, we leverage the intuition that precision in the decision for a hidden unit to activate is less important than carefully tuned analysis weights, which can in part compensate for poorly tuned activation weights. We randomly draw and fix the activation weights $\{\mathbf{W}_l\}$, and then solve for the analysis weights $\mathbf{W_Z}$ using ridge regression, which can be done in a single pass through the dataset. First, each data point $p = 1, \ldots, P$ is expanded into its tensor representation $\mathbf{Z}_L^p$ and then accumulated into the correlation matrices $\mathbf{C_{ZZ}} = \sum_p \mathrm{vec}\left(\mathbf{Z}_L^p\right) \mathrm{vec}\left(\mathbf{Z}_L^p\right)^\intercal$ and $\mathbf{C}_{y\mathbf{Z}} = \sum_p y^p \mathrm{vec}\left(\mathbf{Z}_L^p\right)^\intercal$. After all data points are processed once, the analysis weights are determined as $\mathbf{W_Z} = \mathbf{C}_{y\mathbf{Z}}\left(\mathbf{C_{ZZ}} + \lambda\mathbf{I}\right)^{-1}$ where $\lambda$ is an $L_2$ regularization parameter.

Unlike a standard SS network, which in this setting would only be able to select a linear readout from the top hidden layer to the final classification decision, the TS network offers direct access to entire input vectors, parcelled by the hidden units they activate. In this way, even a linear readout can effectively reach back across layers to the input, implementing a complex function not representable with an SS network with random filters. However, this scheme requires high memory usage, which is on the order of $\mathcal{O}\left(\prod_{l=0}^L n_l^2\right)$ for storing $\mathbf{C_{ZZ}}$, and even higher computation cost[7] for solving $\mathbf{W_Z}$, which makes deep architectures (*i.e.* $L > 1$) impractical. Therefore, this scheme may best suit online learning applications which allow only one-time access to data, but do not require a deep classifier.

### 4.2 Representation Learning via Inverted Backpropagation

The ridge regression learning uses random activation weights and only learns analysis weights. Here we provide a "gradient-based" procedure to learn both weights. Learning the analysis weights (*i.e.* the final linear layer) $\mathbf{W_Z}$ simply requires $\frac{\partial \mathcal{L}_\mathbf{Z}}{\partial \mathbf{W_Z}}$, which is generally easy to compute. However, since the activation weights $\mathbf{W}_l$ in the TS network only appear inside the Heaviside step function $H$ with zero (or undefined) derivative, the gradient $\frac{\partial \mathcal{L}_\mathbf{Z}}{\partial \mathbf{W}_l}$ is also zero. To bypass this, we introduce a sequence of auxiliary variables $\mathbf{A}_l$ defined by $\mathbf{A}_0 = \mathbf{Z}_L$ and the recursion $\mathbf{A}_l = \mathbf{A}_{l-1} \ominus \mathbf{W}_l \in \mathbb{R}^{n_L \times n_{L-1} \times \cdots \times n_l}$. We then derive the pseudo gradient using the proposed inverted backpropagation as

$$\widehat{\frac{\partial \mathcal{L}_\mathbf{Z}}{\partial \mathbf{W}_l}} = \frac{\partial \mathcal{L}_\mathbf{Z}}{\partial \mathbf{A}_0}\left(\frac{\partial \mathbf{A}_1}{\partial \mathbf{A}_0}\right)^\dagger \cdots \left(\frac{\partial \mathbf{A}_l}{\partial \mathbf{A}_{l-1}}\right)^\dagger \frac{\partial \mathbf{A}_l}{\partial \mathbf{W}_l}, \tag{6}$$

where $\dagger$ denotes Moore–Penrose pseudoinverse. Because the $\mathbf{A}_l$'s are related via the linear contraction operator, these derivatives are non-zero and easy to compute. We find this works sufficiently well as a non-zero proxy for $\frac{\partial \mathcal{L}_\mathbf{Z}}{\partial \mathbf{W}_l}$.

Our motivation with this scheme is to "recover" the learning behavior in SS networks. To see this, first note that $\mathbf{A}_L = \mathbf{A}_0 \ominus \mathbf{W}_1 \cdots \ominus \mathbf{W}_L = \mathbf{X}_L$ (see Fig. 3). This reflects the fact that the TS and SS networks are linear once the active set of hidden units is known, such that the order of expansion and contraction steps has no effect on the final output. Hence the linear contraction steps, which alternate with expansion steps in (3), can instead be gathered at the end after all expansion steps. The gradient in the SS network is then

$$\frac{\partial \mathcal{L}_{\mathbf{X}}}{\partial \mathbf{W}_l} = \frac{\partial \mathcal{L}_{\mathbf{X}}}{\partial \mathbf{A}_L} \frac{\partial \mathbf{A}_L}{\partial \mathbf{A}_{L-1}} \cdots \frac{\partial \mathbf{A}_{l+1}}{\partial \mathbf{A}_l} \frac{\partial \mathbf{A}_l}{\partial \mathbf{W}_l} = \underbrace{\frac{\partial \mathcal{L}_{\mathbf{X}}}{\partial \mathbf{A}_L} \frac{\partial \mathbf{A}_L}{\partial \mathbf{A}_{L-1}} \cdots \frac{\partial \mathbf{A}_1}{\partial \mathbf{A}_0}}_{\frac{\partial \mathcal{L}_{\mathbf{X}}}{\partial \mathbf{A}_0}} \left(\frac{\partial \mathbf{A}_1}{\partial \mathbf{A}_0}\right)^{\dagger} \cdots \left(\frac{\partial \mathbf{A}_l}{\partial \mathbf{A}_{l-1}}\right)^{\dagger} \frac{\partial \mathbf{A}_l}{\partial \mathbf{W}_l}. \tag{7}$$

Replacing $\frac{\partial \mathcal{L}_{\mathbf{x}}}{\partial \mathbf{A}_0}$ in (7) with $\frac{\partial \mathcal{L}_{\mathbf{z}}}{\partial \mathbf{A}_0}$, such that the expanded representation may influence the inverted gradient, we recover (6). Compared to one-pass ridge regression, this scheme controls the memory and time complexities at $\mathcal{O}\left(\prod_l n_l\right)$, which makes training of a moderately-sized TS network on modern computing resources feasible. The ability to train activation weights also relaxes the assumption that analysis weights can "clean up" inexact activations caused by using even random weights.

### 4.3 Indirect Representation Learning via Linear Rotation-Compression

Although the inverted backpropagation learning controls memory and time complexities better than the one-pass ridge regression, the exponential growth of a TS network's representation still severely constrains its potential toward being applied in recent deep learning architectures, where network width and depth can easily go beyond, *e.g.*, a thousand. In addition, the success of recent deep learning architectures also heavily depends on the acceleration provided by highly-optimized GPU-enabled libraries, where the operations of the previous learning schemes are mostly unsupported.

To address these 2 concerns, we provide a standard backpropagation-compatible learning algorithm, where we no longer keep separate $\mathbf{X}$ and $\mathbf{Z}$ variables. Instead we define $\mathbf{X}_l = \mathbf{W}_l^* \operatorname{vec}\left(\mathbf{X}_{l-1} \oplus \mathbf{W}_l\right)$, which directly flattens the expanded representation and linearly projects it against $\mathbf{W}_l^* \in \mathbb{R}^{n_l^* \times n_l n_{l-1}}$. In this scheme, even though $\mathbf{W}_l$ still lacks a non-zero gradient, the $\mathbf{W}_{l-1}^*$ of the previous layer can be learned using backpropagation to properly "rotate" $\mathbf{X}_{l-1}$, such that it can be utilized by $\mathbf{W}_l$ and the TS nonlinearity. Therefore, the representation learning here becomes indirect. To simultaneously control the representation size, one can easily let $n_l^* < n_l n_{l-1}$ such that $\mathbf{W}_l^*$ becomes "compressive." Interestingly, we find $n_l^* = n_l$ often works surprisingly well, which suggests linearly compressing the expanded TS representation back to the size of an SS representation can still retain its advantage, and thus is used as the default. This scheme can also be combined with inverted backpropagation if learning $\mathbf{W}_l$ is still desired.

To understand why linear compression does not remove the TS representation power, we note that it is not equivalent to the linear contraction operation $\ominus$, where each tensor-valued unit is down projected independently. Linear compression introduces extra interaction between tensor-valued units. Another way to view the linear compression's role is through kernel analysis as shown in Sec. 3—adding a linear layer does not change the shape of a given TS kernel.

## 5 Experimental Results

Our experiments focus on comparing TS and SS networks with the goal of determining how the TS nonlinearities differ from their SS counterparts. SVMs using SS-ReLU and TS-ReLU kernels are implemented in Matlab based on `libsvm-compact` [22]. TS networks and all 3 learning algorithms in Sec. 4 are implemented in Python based on Numpy's `ndarray` data structure. Both implementations utilize multicore CPU acceleration. In addition, TS networks with only the linear rotation-compression learning are also implemented in Keras, which enjoys much faster GPU acceleration.

We adopt 3 datasets, *viz.* MNIST, CIFAR10 and SVHN2, where we reserve the last 5,000 training images for validation. We also include SVHN2's extra training set (except for SVMs[8]) in the training process, and zero-pad MNIST images such that all datasets have the same spatial resolution—$32 \times 32$.

Table 1: Error rate (%) and run time (×) comparison.

| Error Rate$^{Depth}$ | MNIST<br>One-pass − Asymptotic | CIFAR10<br>One-pass − Asymptotic | SVHN2<br>One-pass − Asymptotic | Time |
|---|---|---|---|---|
| SS SVM | − **1.40**$^5$ | − **43.18**$^7$ | − 21.60$^1$ | 1.0 |
| TS SVM | − **1.40**$^3$ | − 43.60$^2$ | − **20.38**$^1$ | 2.1 |
| SS MLP | 16.34$^2$ − 2.36$^3$ | 66.41$^1$ − 46.91$^2$ | 30.24$^3$ − **12.20**$^3$ | 1.0 |
| TS MLP *RR* | **2.99**$^1$ − | **47.71**$^1$ − | 27.11$^1$ − | 156.2 |
| TS MLP *LRC* | 3.33$^2$ − **2.06**$^2$ | 55.69$^1$ − 46.87$^2$ | 20.42$^2$ − 12.58$^3$ | 11.7 |
| TS MLP *IBP-LRC* | 3.33$^1$ − 2.33$^1$ | 55.69$^1$ − **45.86**$^2$ | **20.20**$^2$ − 12.63$^3$ | 17.4 |
| SS CNN | 43.74$^{3+1}$ − 1.08$^{4+2}$ | 74.84$^{3+3}$ − 26.73$^{5+2}$ | 13.69$^{7+1}$ − **4.96**$^{6+1}$ | 1.0 |
| TS CNN *LRC* | **3.85**$^{5+3}$ − **0.86**$^{6+2}$ | **54.40**$^{3+3}$ − **25.74**$^{8+3}$ | **9.13**$^{7+3}$ − 5.06$^{6+3}$ | 2.0 |

*RR* = One-Pass Ridge Regression, *LRC* = Linear Rotation-Compression, *IBP* = Inverted Backpropagation.

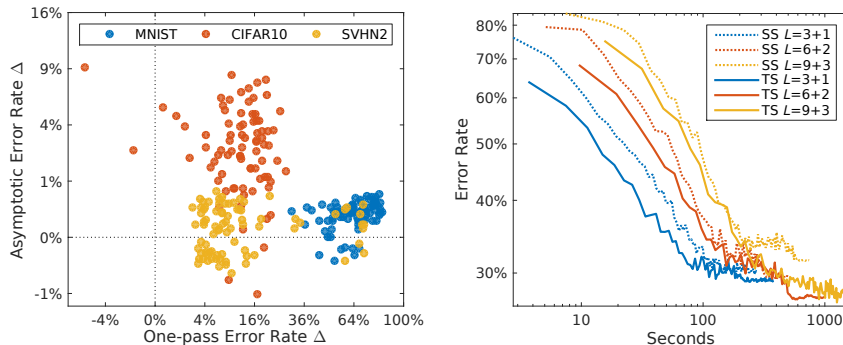

Figure 4: Comparison of SS CNN and TS CNN *LRC* models. (Left) Each dot's coordinate indicates the differences of one-pass and asymptotic error rates between one pair of SS CNN and TS CNN *LRC* models sharing the same hyperparameters. The first quadrant shows where the TS CNN *LRC* is better in both errors. (Right) Validation error rates *v.s.* training time on CIFAR10 from the shallower, intermediate and deeper models.

For SVMs, we grid search for both kernels with depth from 1 to 10, $C$ from 1 to $1,000$, and PCA dimension reduction of the images to 32, 64, 128, 256, or no reduction. For SS and TS networks with fully-connected (*i.e.* MLP) architectures, we grid search for depth from 1 to 3 and width (including PCA of the input) from 32 to 256 based on our Python implementation. For SS and TS networks with convolutional (*i.e.* CNN) architectures, we adopt VGG-style [15] convolutional layers with 3 standard SS convolution-max pooling blocks,[9] where each block can have up to three $3 \times 3$ convolutions, plus 1 to 3 fully-connected SS or TS layers of fixed width 256. CNN experiments are based on our Keras implementation. For all MLPs and CNNs, we universally use SGD with learning rate $10^{-3}$, momentum 0.9, $L_2$ weight decay $10^{-3}$ and batch size 128 to reduce the grid search complexity by focusing on architectural hyperparameters. All networks are trained for 100 epochs on MNIST and CIFAR10, and 20 epochs on SVHN2, without data augmentation. The source code and scripts for reproducing our experiments are available at `https://github.com/coxlab/tsnet`.

Table 1 summarizes our experimental results, including both one-pass (*i.e.* first-epoch) and asymptotic (*i.e.* all-epoch) error rates and the corresponding depths (for CNNs, convolutional and fully-connected layers are listed separately). The TS nonlinearities perform better in almost all categories, confirming our theoretical insights in Sec. 2.3—the cross-layer analysis (as evidenced by their low error rates after only one epoch of training), the error-correcting analysis (on MNIST and CIFAR10, for instance, the one-pass error rates of TS MLP *RR* using fixed random activation are close to the asymptotic error rates of TS MLP *LRC* and *IBP-LRC* with trained activation), and the fine-grained analysis (the TS networks in general achieve better asymptotic error rates than their SS counterparts).

| Backpropagation (SS MLP) | Inverted Backpropagation (TS MLP *IBP*) |

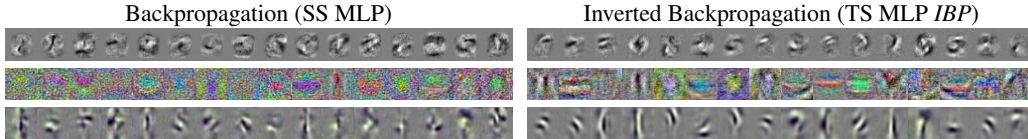

Figure 5: Visualization of filters learned on (Top) MNIST, (Middle) CIFAR10 and (Bottom) SVHN2.

To further demonstrate how using TS nonlinearities affects the distribution of performance across different architectures (here, mainly depth), we plot the performance gains (*viz.* one-pass and asymptotic error rates) introduced by using the TS nonlinearities on all CNN variants in Fig. 4. The fact that most dots are in the first quadrant (and none in the third quadrant) suggests the TS nonlinearities are predominantly beneficial. Also, to ease the concern that the TS networks' higher complexity may simply consume their advantage on actual run time, we also provide examples of learning progress (*i.e.* validation error rate) over run time in Fig. 4. The results suggest that even our unoptimized TS network implementation can still provide sizable gains in learning speed.

Finally, to verify the effectiveness of inverted backpropagation in learning useful activation filters even without the actual gradient, we train single-hidden-layer SS and TS MLPs with 16 hidden units each (without using PCA dimension reduction of the input) and visualize the learned filters in Fig. 5. The results suggest inverted backpropagation functions equally well.

# 6   Discussion

**Why do TS networks learn quickly?** In general, the TS network sidesteps the vanishing gradient problem as it skips the long chain of linear contractions against the analysis weights (*i.e.* the auxiliary pathway in Fig. 3). Its linear readout has direct access to the full input vector, which is switched to different parts of the highly expressive expanded representation. This directly accelerates learning. Also, a well-flowing gradient confers benefits beyond the TS layers—*e.g.* SS layers placed before TS layers also learn faster since the TS layers "self-organize" rapidly, permitting useful error signals to flow to the lower layers faster.[10] Lastly, when using the inverted backpropagation or linear rotation-compression learning, although $\{\mathbf{W}_l\}$ or $\{\mathbf{W}_l^*\}$ do not learn as fast as $\mathbf{W_Z}$, and may still be quite random in the first few epochs, the error-correcting nature of $\mathbf{W_Z}$ can still compensate for the learning progress.

**Challenges toward deeper TS networks.** As shown in Fig. 2, the equivalent kernels of deeper TS networks can be extremely sharp and discriminative, which unavoidably hurts invariant recognition of dissimilar examples. This may explain why we find having TS nonlinearities in only higher (instead of all) layers works better, since the lower SS layers can form invariant representations for the higher TS layers to classify. To remedy this, we may need to consider other types of regularization for $\mathbf{W_Z}$ (instead of $L_2$) or other smoothing techniques [25, 26].

**Future work.** Our main future direction is to improve the TS network's scalability, which may require more parallelism (*e.g.* multi-GPU processing) and more customization (*e.g.* GPU kernels utilizing the sparsity of TS representations), with preferably more memory storage/bandwidth (*e.g.* GPUs using 3D-stacked memory). With improved scalability, we also plan to further verify the TS nonlinearity's efficiency in state-of-the-art architectures [27, 9, 18], which are still computationally prohibitive with our current implementation.

**Acknowledgments**

We would like to thank James Fitzgerald, Mien "Brabeeba" Wang, Scott Linderman, and Yu Hu for fruitful discussions. We also thank the anonymous reviewers for their valuable comments. This work was supported by NSF (IIS 1409097), IARPA (contract D16PC00002), and the Swartz Foundation.

## Footnotes

[1]For convolutional networks using max pooling, the convolutional-window-sized input patch winning the max pooling is copied. In other words, different nonlinearities only change the way the input is switched.

[2]It is in spirit similar to models with skip connections to the output [17, 18], although not exactly reducible.

[3]Therefore TS networks are also universal function approximators [19].

[4] This proof is succinct using a geometric view, while a longer proof can be found in the Supplementary Material. As the kernel is directly defined as a dot product between feature vectors, it is naturally a valid kernel.

[5] Interestingly, a similar kernel is also observed by [21] for models with explicit skip connections.

[6] We write (4) and $k_l^{\text{SS}}$ differently from [20] for cleaner comparisons against TS-ReLU kernels. However they are numerically unstable expressions and are not used in our experiments to replace the original ones in [20].

[7]Nonetheless this is a one-time cost and still can be advantageous over other slowly converging algorithms.

[8]Due to the prohibitive kernel matrix size, as SVMs here can only be solved in the dual form.

[9]This decision mainly is to accelerate the experimental process, since TS convolution runs much slower, but we also observe that TS nonlinearities in lower layers are not always helpful. See later for more discussion.

[10]This is a crucial aspect of gradient descent dynamics in layered structures, which behave like a chain—the weakest link must change first [23, 24].

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
