[Supplementary Material · tsnet_sup.pdf]

# Tensor Switching Networks: Supplementary Material

**Chuan-Yung Tsai**[*], **Andrew Saxe**[*], **David Cox**
Center for Brain Science, Harvard University, Cambridge, MA 02138
{chuanyungtsai,asaxe,davidcox}@fas.harvard.edu

## Alternative Derivation of TS-ReLU Kernel

Given $\mathbf{x}, \mathbf{y} \in \mathbb{R}^{n_0}$, we wish to derive $\bar{k}^{\text{TS}}$ in

$$k_1^{\text{TS}}(\mathbf{x}, \mathbf{y}) = 2\,\mathbb{E}\left[H(\mathbf{w}^{\mathsf{T}}\mathbf{x})\,H(\mathbf{w}^{\mathsf{T}}\mathbf{y})\right]\mathbf{x}^{\mathsf{T}}\mathbf{y} = \underbrace{2\,P(\mathbf{w}^{\mathsf{T}}\mathbf{x} > 0 \text{ and } \mathbf{w}^{\mathsf{T}}\mathbf{y} > 0)}_{\bar{k}^{\text{TS}}}\mathbf{x}^{\mathsf{T}}\mathbf{y},$$

where $\mathbf{w} \sim \mathcal{N}\left(\mathbf{0}, \sigma^2\mathbf{I}\right)$. To achieve this goal, we define

$$\begin{bmatrix} z_1 \\ z_2 \end{bmatrix} = \underbrace{\begin{bmatrix} \mathbf{x}^{\mathsf{T}}/\sigma\,\|\mathbf{x}\| \\ \mathbf{y}^{\mathsf{T}}/\sigma\,\|\mathbf{y}\| \end{bmatrix}}_{\mathbf{L}}\mathbf{w} \sim \mathcal{N}\left(\mathbf{0}, \underbrace{\begin{bmatrix} 1 & \cos\theta \\ \cos\theta & 1 \end{bmatrix}}_{\mathbf{L}(\sigma^2\mathbf{I})\mathbf{L}^{\mathsf{T}}}\right). \tag{1}$$

Then we have

$$\bar{k}^{\text{TS}} = 2\,P(\mathbf{w}^{\mathsf{T}}\mathbf{x} > 0 \text{ and } \mathbf{w}^{\mathsf{T}}\mathbf{y} > 0)$$

$$= 2\,P\left(\frac{\mathbf{w}^{\mathsf{T}}\mathbf{x}}{\sigma\,\|\mathbf{x}\|} > 0 \text{ and } \frac{\mathbf{w}^{\mathsf{T}}\mathbf{y}}{\sigma\,\|\mathbf{y}\|} > 0\right)$$

$$= 2\,P(z_1 > 0 \text{ and } z_2 > 0)$$

$$= 2\int_0^\infty \int_0^\infty \frac{1}{2\pi\sqrt{1-\cos^2\theta}}\exp\left(-\frac{z_1^2 - 2z_1 z_2 \cos\theta + z_2^2}{2(1-\cos^2\theta)}\right)dz_1 dz_2 \quad \text{Using PDF of (1)}$$

$$= \frac{1}{\pi\sin\theta}\int_0^{\frac{\pi}{2}}\int_0^\infty r\exp\left(-r^2\frac{1-\cos\theta\sin 2\phi}{2\sin^2\theta}\right)dr d\phi \qquad\qquad \text{Polar Coordinates}$$

$$= \frac{1}{\pi\sin\theta}\int_0^{\frac{\pi}{2}}\left(-\frac{1}{2a}\exp\left(-r^2 a\right)\Big|_{r=0}^\infty\right)d\phi \qquad\qquad a = \frac{1-\cos\theta\sin 2\phi}{2\sin^2\theta}$$

$$= \frac{1}{\pi\sin\theta}\int_0^{\frac{\pi}{2}}\frac{1}{2a}d\phi$$

$$= \frac{1}{\pi}\sin\theta\int_0^{\frac{\pi}{2}}\frac{1}{1-\cos\theta\sin 2\phi}d\phi \qquad\qquad \text{Special Case of (A.3) of [1]}$$

$$= \frac{1}{\pi}\sin\theta\left(\frac{\pi-\theta}{\sin\theta}\right) \qquad\qquad \text{Following (A.6) of [1]}$$

$$= 1 - \frac{\theta}{\pi}.$$

Thus, $k_1^{\text{TS}}(\mathbf{x}, \mathbf{y}) = \left(1 - \frac{\theta}{\pi}\right)\mathbf{x}^{\mathsf{T}}\mathbf{y}$.

---

[*]Equal contribution.

# References

[1] Y. Cho and L. Saul, "Large-Margin Classification in Infinite Neural Networks," *Neural Computation*, 2010.