[Reviews · NeurIPS 2016]

Reviewer 1

Summary

The paper proposes a new network architecture that, like ReLU networks, is conditionally linear. The novelty in the proposed architecture is that the weights that determine which units are active (called activation weights in the paper) are decoupled from the weights that map the input to the output (called analysis weights in the paper). In the proposed architecture, active units copy their full input to the output, while inactive units output zeros. As a consequence of this design, the output of each layer is a tensor, and each successive layer in the network increases the order of the tensor by one. The analysis weights are applied only to the final output. Three different algorithms for training these tensor switching networks are described in the paper. The first derives a kernel function that is equivalent to one or more tensor switching layers that are infinitely wide and have activation weights drawn from a standard normal distribution. Such kernels may be used with small data sets to learn SVMs, and comparisons between the tensor switching kernel and the arccosine kernel (which is the equivalent kernel for deep ReLU networks) provide insights into the differences between tensor switching and ReLU networks. The second training algorithm draws the activation weights randomly from a given distribution and then uses regularized ridge regression to learn the analysis weights using only a single pass over the training data. The third training algorithm introduces introduces a set of auxiliary variables such that the derivatives of the loss with respect to the auxiliary variables serve as proxies for the derivatives of the loss with respect to the unit activation weights, which are zero or undefined everywhere. Experiments on variations of MNIST show that the combination of tensor switching networks and the second training algorithm provide the best performance when learning is done in a single pass over the training data.

Qualitative Assessment

This is a creative piece of work, and the provision of three different learning algorithms that work and help provide insights into the proposed tensor switching architecture is impressive. I have only a few minor comments that, if addressed, would improve the paper. Figures 2a, b, and c would be clearer if the x axes were labeled in terms of fractions of pi. Page 7, lines 242-243: "In terms of running time, the SS network enjoys a roughly 8× improvement in wall-clock training speed over the SS network for the 1 layer MLP." Should this be "...the RR network enjoys..."? Page 7, lines 247-248: "the TS net architecture, [which] attains much lower asymptotic training error than the SS net but similar test errors" -- basically, TS networks overfit more than the SS networks. But this isn't a surprise, is it? The TS networks have far more parameters than the SS networks with the same architecture. One way of looking at this is that the TS networks trade off a large number of parameters for ease and speed of training. How were the sizes of the MLPs in the experiments selected? The one-pass training algorithm with ridge regression is somewhat reminiscent of one-pass training using randomized decision trees. The theory developed for randomized decision trees might provide a useful path for theoretical analysis of TS networks with random activation weights.

Confidence in this Review

2-Confident (read it all; understood it all reasonably well)


Reviewer 2

Summary

The paper proposes a new activation function similar to Relu in the sense that only works in the positive range of values. The way it works this new activation function is by-passing the complete previous layer activation (Tensor). The authors derive a equivalent kernel, an inverted back propagation learning and propose experiments under the online learning framework.

Qualitative Assessment

This paper is a mixture of things. First of all i do not see any motivation in order to by-pass all the previous layer activations (the same) to next layer, perhaps in order to provide a kind of networks combination in later layers (regular ones). Then i do not see any kind of motivation to introduce an equivalent kernel and use SVM's. The experiments with SVM's used a non-standard partition of MNIST, therefore is difficult to compare with state of the art, is this a tactic? Then the authors proposed a kind of backdrop algorithm (more interesting than the kernel) but finally the paper turns to the online framework where experiments, again, are not comparable with state of the art online learning datasets. This idea could have some interest but authors should use standar evaluation with standard corpus.

Confidence in this Review

2-Confident (read it all; understood it all reasonably well)


Reviewer 3

Summary

This paper introduces a novel neural network architecture called Tensor Switching Network (TS Net). This network exploits the structure of the ReLU activation function further by generalizing it such that each hidden unit represents a vector of activities (inactive – zeros, active – copy of their input). TS Net gives a more expressive model. This paper described several algorithms: Infinite-width multilayer Tensor Switching, one-pass learning algorithm and inverted backpropagation learning for the TS Net. In short, TS Net is a generalization of ReLU networks that allows each hidden cell to carry a vector of activity (active or inactive).

Qualitative Assessment

The idea presented in this paper is novel but there are some issues should be addressed in details in the paper. Since TS Ridge Regression only computes once (one epoch), its results should not be plotted together with others (multi-epochs learning) in Fig. 4. Having straight line in that graph makes confusion for readers. As my understanding, TS networks proposed in this paper are a highly expressive model for their size and depth but the memory requirement in this method limits to explore a deeper network. Indeed, only one or two layers settings are reported for RR and IBP in Table 2. Table 2 should include the training time for each method (RR, IBP and SS) for better comparisons. The organization of this paper is fine overall except section 3.2. It should be incorporated into section 6 experimental results.

Confidence in this Review

2-Confident (read it all; understood it all reasonably well)


Reviewer 4

Summary

The paper deals with analysis of the ReLU based Neural Networks. Furthermore it exploits linear structure of the network when it is a-priory known which ReLUs are on and off. A tensor product formalism is developed to this end. Under assumption of the Gaussian weight statistics a kernel for a shallow network is derived and then generalized to a deep network case (arccos iterated kernel)

Qualitative Assessment

This paper entertain an interesting possibility of using tensor product representation to separate ReLU network into linear and non-linear switching part and treat those parts separately. This separation cannot be completely achieved, for terms like W*X are present both inside the Heaviside step function \Theta(WX) and outside of it leading to non-trivial coupling of activations and the poly-linear terms (e.g. see [9], eqs (1),(3)). I think the authors should be encouraged to continue the work and produce a more detailed journal article on this subject with more thorough derivations of the kernels and their compositions.

Confidence in this Review

2-Confident (read it all; understood it all reasonably well)


Reviewer 5

Summary

The paper proposes a generalized form from ReLU network to tensor switching network where the hidden units pass the input vector when activated, instead a single scalar in the ReLU net. The TS net is examined in three aspects (infinite kernels, one-pass learning, inverted BP) to show the properties and advantages.

Qualitative Assessment

Pros: 1. The paper is well written with solid reasoning and clear arguments. It explains well how the TS net descends from the standard ReLU network and lists three highlight differences in Section 2. Cons: 1. Novelty is not that strong for a NIPS poster. The TS network is derived, in essence, from the tensor decomposition in the ReLU non-linearity formulation and the authors give a new explanation and propose a new matrix representation Z between the input X_0 to output X_1, which decouples activation from ‘analysis’. While I admit the first two advantages are obvious (ln 91, 97), the third advantage of the TS net does not hold. There’s no analysis filter in the deep SS network and parameters need not to share in some cases (like fully connected layers). 2. Inaccurate expressions. Ln 114 states the following three methods are to ‘learn the parameters’ of a TS network, which is contradictory to Ln 121, Ln 197. 3. About the first two algorithms that exploit TS net’s linearity. (1) The paper derives some kernels to equal the TS net with infinite hidden units. From the experiments on MNIST we can see a little margin in performance between TS and SS (in percent) and the TS net prefers shallower depths (=2) otherwise the error will increase. This is probably due to the enlarged parameters in the analysis filters W_z. Therefore, I am wondering if the proposed method can fit into really deep network, say just 10 layers although they claim so theoretically in Section 3.1. (2) The abstract and intro claims that the one-pass learning visits each training sample only once and ‘hence can scale to very large datasets’. However, there’s no experiments on relatively ‘large’ datasets to prove this. Also, the main claim in Section 4 is to prove the effectiveness of TS despite poorly tuned activation weights, I see no relation between one-sample learning and scalability to large datasets. 4. Experiments are not enough to prove the effectiveness of the TS network. For example, the visualization of Figure 5 does not show a superior feature representation learning of TS over SS; the performance in Table 2 is not convincing as it only has shallow networks (3 layers) and compares only to the SS design with a little margin in performance, and in some cases the SS is better. Also as mentioned earlier, the paper keeps selling TS’s adaptability to large or real-time data (ln 261) and yet we see not experiment regarding this aspect. 5. No clear related work part. No need to create a separate section but it would be better to input them together in the intro or somewhere before the experiments. Currently it’s spread all over the paper, ln 46, ln 171, etc. It’s hard for the audience to understand what is really novel or different from previous works.

Confidence in this Review

2-Confident (read it all; understood it all reasonably well)


Reviewer 6

Summary

This paper proposes a novel linear architecture to build deep neural networks, namely Tensor Switching Network (TS Net), as an alternative to RELU. This architecture has the advantage that after just one pass through the training data it performs much better than RELU. However, after more passing (i.e. training epochs) TS Net does not improve its performance, while the standard RELU architecture improves considerably and outperform drastically TS Net. Thus, as far I can see (suggested also by the authors), the only practical applicability of TS Net is in online learning settings.

Qualitative Assessment

Overall the paper is well written and clear. While TS Net seems a promising solution to adapt deep networks to online learning settings, it has a clear scalability problem, in my opinion. Even in online learning settings, the fact that TS Net RR (with ridge regression learning) needs for a relatively small dataset (i.e. MNIST variants), at least 200 times more memory than RELU and at least 6 times more computational time per training epochs makes it a bit unfeasible. If you imagine that in online settings, one would like to do the computations fast and (maybe) on low resources devices, this does not sound very appealing. The other case, TS Net IBP (with inverted back propagation learning) needs (indeed) less memory or computational resources, but also needs more training epochs to achieve the best results which are still far below the ones obtained by the RELU architecture. Thus, TS Net IBP seems to be not useful in either case, online or offline settings. Besides that, the experimental section lacks some variability. The experiments are performed just on variants of the MNIST dataset. Especially because TS Net has some downsides as discuss above, it would be interesting to see how it performs on different (maybe bigger) datasets coming from different domains. Have you test such situations? To conclude, I really like the idea of the authors, and I would encourage them to search for faster training methods and for some ways to reduce the exponential grows of the number of parameters. Maybe some matrix factorization would help?

Confidence in this Review

2-Confident (read it all; understood it all reasonably well)